# Impact of School Closures, Precipitated by COVID-19, on Weight and Weight-Related Risk Factors among Schoolteachers: A Cross-Sectional Study

**DOI:** 10.3390/nu13082723

**Published:** 2021-08-07

**Authors:** Jill R. Silverman, Branden Z. Wang

**Affiliations:** 1Department of Nutrition Science and Wellness, Farmingdale State College, 2350 Broadhollow Road, Farmingdale, NY 11735, USA; 2Institute of Human Nutrition, Columbia University Irving Medical Center, New York, NY 10027, USA; bw2642@cumc.columbia.edu

**Keywords:** coronavirus, exercise, emotional eating, pandemic, quarantine, questionnaire, sedentariness, New York

## Abstract

The school closures, precipitated by the COVID-19 pandemic, required teachers to convert their entire classroom curricula to online formats, taught from home. This shift to a more sedentary teaching environment, coupled with the stresses related to the pandemic, may correlate with weight gain. In total, 52% of study participants reported weight gain, with a higher prevalence observed among kindergarten and elementary school teachers when compared to high school teachers (*p* < 0.05). Deviations in physical activity, emotional eating, and dietary patterns were assessed among 129 teachers (using the Leisure Time Exercise Questionnaire, the Dutch Eating Behavioral Questionnaire, and a short-form Food Frequency Questionnaire, respectively) to uncover possible associations with the observed weight gain. Increases in sedentariness (*p* < 0.005), emotional eating (*p* < 0.001), the consumption of potatoes, fries, breads, cheese, cake (*p* < 0.05), chips, candy, ice-cream, and soft drinks (*p* < 0.005) were all positively correlated with weight gain. Decreases in exercise frequency (*p* < 0.001), and the consumption of fruits (*p* < 0.05) and beans (*p* < 0.005), were also positively correlated with weight gain. Weight gain, observed among teachers during school closures, was associated with changes in diet, emotional eating and physical activity.

## 1. Introduction

The coronavirus pandemic has caused significant disruption in the lifestyles of Americans. At the beginning of March 2020, there was a huge increase in the number of confirmed cases, hospitalizations, and deaths resulting from the virus [1], precipitating severe restrictions nationwide [2]. In New York state, mandatory shelter-in-place guidelines were implemented, and people were only allowed to leave their homes for food or medical reasons. As with many other occupations, schoolteachers were required to make the rapid and unplanned move to working from home [3]. For teachers, who spend most of their days standing in classrooms interacting with their students, this meant adapting lesson plans to an online format, significantly increasing the amount of time spent at home in front of their computer screens [4].

The impact of the mandatory shelter-in-place restrictions on changes in weight, dietary habits, and physical activity have been demonstrated by several studies [5,6,7,8]. In addition, studies have shown that an increase in unstructured time, a result of the COVID-19 school closures, can result in an increase in weight gain-related risk factors among children [9,10]. However, the effect of these school closures on teachers has not been determined. It was hypothesized that the unstructured time and stress associated with the shift to a virtual classroom and working from home, might lead to deviations in food intake, activity levels, and emotional eating, and that these changes may correlate to weight gain among teachers.

Time spent sitting is greatly associated with increased rates of obesity [11]. During a traditional school year, teachers spend, on average, 12–16 h a day devoted to the direct instruction of students, curriculum development, and administrative duties [12]. With the closing of schools and switch to teaching virtually from home, teachers were no longer actively moving within the school building. Therefore, the change in the teaching environment may have resulted in an increase in sedentary activities and possible weight gain.

Furthermore, the increase in stress due to the pandemic and subsequent school closings may have triggered emotional eating; the use of food for comfort when experiencing negative emotions. Increased levels of stress have been reported during the pandemic [13,14,15]. Stressors associated with the pandemic include fear of infection, frustration, boredom, isolation, home schooling of children, loss of unemployment, and financial loss [16,17,18,19,20,21,22,23]. Previous studies have demonstrated that stress is correlated with deviations in emotional eating, eating patterns, and food choices [24,25,26,27,28]. Furthermore, a positive association between stress and an increased appetite for palatable, calorically dense foods has also been reported [29,30] with a preference for “comfort foods” (foods high in calories, sugar, and fat) [31,32,33]. It has been observed that emotional eaters tend to ingest greater amounts of sweet, fatty, and salty foods during difficult times [34,35]. The cause of these changes in food consumption may be due to alterations in cortisol or satiety hormones, which are negatively affected during periods of chronic stress [36,37,38,39]. Therefore, an observed increase in the consumption of these calorically dense “comfort” foods could also be associated with weight gain among teachers.

Not only has stress been associated with emotional eating, but several studies have demonstrated that stress is correlated with a decrease in physical activity and an increase in sedentariness. Further, it was indicated that both objective and subjective indicators of stress were associated with the observed reduction in physical activity [40,41,42].

Cumulatively, all of these factors have the potential to induce weight gain. Therefore, the purpose of this study was to determine if the mandatory school closures (16 March–26 June 2020), a result of the COVID-19 pandemic, were associated with weight gain among schoolteachers, and if so, did the weight gain correlate with changes in food consumption, emotional eating or physical activity.

## 2. Materials and Methods

A cross-sectional study was conducted on public-school teachers in Long Island, a suburban area in the east of New York City. Teachers were recruited in June and July 2020, via social media (Long Island teacher Facebook pages) and mass emails, to take part in this study. Informed consent was obtained from all 129 subjects prior to their access to the survey. Inclusion criteria were: (a) full-time teacher in a public school in Long Island, NY, during the 2019–2020 school year (b) not pregnant or lactating (c) have internet access. The study was completed in accordance with the IRB of Farmingdale State College, NY, USA.

### 2.1. Study Design

Once subjects voluntarily agreed to participate in the study, they received the link to a Qualtrics (Seattle, WA, USA) survey consisting of 52 questions. The descriptive information collected from the subjects included age, gender, marital status, employment status, duration of years teaching, income, ethnicity, and self-reported height and weight. Participants weight in pounds was used to determine changes in body weight. Subjects were also asked how many total people, and how many people under the age of 18 years, were residing in the home during the 3-month quarantine period, and which grade they were teaching during the school closures. In most school districts on Long Island, kindergarten classrooms are in the same building as elementary school grades—first through fifth—and will be referred to as K-5. Middle school (MS) encompasses grades 6 through 8, and high school (HS) grades 9 through 12. Both MS and HS are, customarily, housed within their own separate school buildings. A short-form food-frequency questionnaire (SFFFQ) was used to assess food intake. The participants were asked to report the frequency of consumption of alcoholic and non-alcoholic beverages, fruits, vegetables, starches, fiber-rich foods, high-fat and high-sugar foods, meats, and dairy products prior to (before 16 March 2020) and during (up until 26 June 2020) school closures. Frequency of consumption was measured using a scale ranging from less than once a month to 2 or more times per day. SFFFQs have been found to be reliable and validated for measuring food consumption [43]. The Leisure-Time Exercise Questionnaire (LTEQ) was used to assess changes in the frequency and intensity of physical activity over the 3-month period. Participants were asked, on average, how many hours per week they engaged in the following exercises: strenuous exercise (heart beats rapidly), moderate exercise (not exhausting), mild exercise (minimal effort). Examples of exercises from each of the 3 categories were provided. The LTEQ has been found to be reliable and valid [44]. The frequency of emotional eating was measured using the Dutch Eating Behavior Questionnaire (DEBQ). The DEBQ contains 13 questions that could determine whether certain emotional states trigger eating (emotional eating). The participants were asked to report on the frequency of their emotional eating prior to and during school closures using a 5-answer scale ranging from “never” to “very often.” The DEBQ has been tested for reliability and validity [45].

### 2.2. Statistics

Text answers from the SFFFQ, LTEQ, and DEBQ were converted to a number, based on the indicated frequency. Nearly all data, converted and non-converted, were non-parametric and not normally distributed. Therefore, Wilcoxon signed-rank tests were used to measure any changes in dietary habits, physical activity, and emotional eating during school closures. Spearman’s rank-order correlation test was used to find correlation. One-way ANOVA was used for parametric data. Dunn’s Kruskal–Wallis test was used to compare the differences between subgroups nonparametrically. Median and interquartile ranges are reported for the comparisons with significant *p*-value. All statistical analyses were performed using R. Significance for all tests was set as *p* < 0.05.

## 3. Results

### 3.1. Baseline Characteristics

Table 1 depicts the baseline characteristics of the participants, separated by grade taught during the 2019–2020 school year and the demographic questions that were included in the survey. As can be seen in the table, approximately 51% of the respondents taught in K-5 classrooms during the shelter-in-place period. Totals of 26% and 23% taught in middle school and high school classrooms, respectively. The majority of respondents were white females, similar to the national demographics for teachers (80% white and 77% female) [46].

### 3.2. Weight Change

To determine if the school closures, and complete shift from the school classroom to the home classroom, were associated with weight change among the teachers, we compared the mean body weight prior to (M = 157.09, SD = 33.5) and during (M = 158.8, SD = 33.6) the school closures. As depicted in Figure 1A, there was no significant change in body weight observed among the 129 participants. However, when we separated the teachers by grade taught (Figure 1B), the subgroup analysis revealed that participants who taught K-5 (Mdn = 4, IQR = −2–8) had a significant gain in weight compared to participants who taught HS (Mdn = −2, IQR = −7–5) (Kruskal–Wallis statistic = 6.781, *p* = 0.03).

To further explore these changes in weight, we separated the teachers into five groups: lost more than 10 pounds; gained more than 10 pounds; lost less than 10 pounds; gained less than 10 pounds; no change in weight (12%, 18%, 27%, 34%, and 9%, respectively). When we analyzed the amount of weight gained and weight lost by teachers according to grade taught, 23% of K-5 teachers reported a weight gain of 10 or more pounds compared to 18% and 7% of MS and HS teachers, respectively. Further, 12% of MS and 24% of HS teachers lost 10 or more pounds, whereas only 6% of K-5 teachers reported this amount of weight loss.

### 3.3. Lifestyle Factors Associated with Weight Gain

As previously mentioned, increases in the consumption of “comfort” foods and emotional eating have been reported during times of stress. Figure 2A illustrates a significant increase in the consumption of alcohol, soft drinks, and calorically dense foods (cereal, chips, potatoes, fries, candy, ice cream, cakes, rice, pizza) among the participants during the school closures. In contrast, considerable reductions in the consumption of raw vegetables and whole-grain bread products, were observed. Regarding emotional eating there were significant increases in food consumption for all 13 categories of emotional eating measured by the Dutch Eating Behavior Questionnaire (DEBQ) (Figure 2B).

In order to determine if these variations in food intake and emotional eating were associated with changes in weight, we conducted a Spearman’s rank-order correlation test. Table 2 illustrates that an increase in the consumption of calorically dense foods (chips, potatoes, fries, bread products, cheese, candy, ice cream, cake, and soft drinks) was positively correlated with weight gain. In contrast, increased consumption of fresh fruits and beans was negatively correlated with weight gain. Further, an increase in emotional eating, regardless of the cause, was positively associated with weight gain, with a particularly strong correlation seen between weight gain and eating when irritated. Lastly, Table 2 also shows that an increase in the frequency of exercise at any intensity (strenuous, moderate, mild/light) was negatively correlated with weight gain, whereas sedentary activities were positively correlated with weight gain.

### 3.4. Factors Affecting Weight by Pounds Lost or Gained

To further determine the contribution of the observed changes in food intake and physical activity to weight gained, we performed nonparametric Dunn tests. As presented in Table 3, among participants who gained more than 10 pounds, we observed a significant increase in the consumption of chips, potatoes, fries, other bread products, candies, ice cream, cakes, and soft drinks when compared to the participants who lost more than 10 pounds. A significant increase in the consumption of chips, potatoes, other bread products, candy, ice cream, and soft drinks among those who gained more than 10 pounds when compared to teachers who lost less than 10 pounds was observed, as well. Regarding the contribution of physical activity to weight change, among teachers who lost any amount of weight there was a significant increase in the frequency of strenuous exercise when compared to the no change in weight and weight gain groups. Participants who gained more than 10 pounds reported a significant decrease in the frequency of both strenuous and moderate exercise, but a significant increase in mild/light exercise and sedentary activities when compared to teachers that lost more than 10 pounds.

### 3.5. Food Consumption among Teachers of Different Grade Levels

As noted above, we revealed a significant weight gain among the K-5 teachers, with a likelihood of 1.51 (95%CI: 1.06–2.14, *p*-value = 0.02). Further, when a chi-square test of independence was performed to determine if there was a correlation between grade taught and weight gain, we observed a significant correlation between these variables, χ^2^ (1, *n* = 129) = 4.81, *p* = 0.03. To further elucidate which factors contributed to the significant weight gain observed among K-5 teachers, we performed pairwise comparisons using Dunn’s all-pairs tests on food intake. When evaluating the data from the SFFFQ separated by grade taught (Table 4), we noticed a significant increase in the consumption of chips and meat products among K-5 teachers when compared to HS teachers (*p* = 0.03 and 0.02, respectively). In addition, the reported consumption of fruits was significantly reduced among K-5 teachers when compared to both MS and HS teachers (*p* = 0.008 and =0.03, respectively). Further, a considerable reduction in the intake of raw vegetables was observed among the K-5 teachers when compared to MS teachers (*p* = 0.03).

### 3.6. Eating Patterns among Teachers of Different Grade Levels

Lastly, we examined whether there were any specific changes in eating patterns among the teachers according to grade taught during the school closures. We observed a considerable increase in the frequency of eating while working in K-5 teachers (Mdn = 9, IQR = 0–29) when compared to both the MS and HS teachers (Mdn = 0 and 0, IQR = −17–12.8 and −17–8, *p* = 0.001 and 0.003, respectively). Kruskal–Wallis χ^2^ (2, *n* = 129) = 14.38, *p* = 0.0008. In addition, a significant increase in the consumption of foods consumed when things had gone wrong was observed among K-5 teachers (Mdn = 0, IQR = 0–1), in contrast to MS and HS teachers (Mdn = 0 and 0, IQR = 0–0 and 0–0, *p* = 0.03 and 0.01, respectively). Kruskal–Wallis χ^2^ (2, *n* = 129) = 8.79, *p* = 0.01.

## 4. Discussion

The COVID-19 pandemic put enormous stress on the nation; economically, emotionally, and physiologically. Although the long-term health effects of the pandemic, and subsequent quarantine, are yet to be elucidated, we were able to identify some of the short-term effects on weight, overall eating habits and frequency of emotional eating and physical activity among kindergarten to 12th grade schoolteachers in Long Island, NY. Although no significant change in weight was observed when comparing all 129 teachers during the three-month school closures, subgroup analysis demonstrated a significant weight gain among K-5 teachers when compared to non-K-5 teachers. Specifically, we noted that 23% of K-5 teachers gained 10 or more pounds compared to 18% and 7% of MS and HS teachers, respectively. Further, 12% of MS and 24% of HS teachers lost 10 or more pounds, whereas only 6% of K-5 teachers reported losing this amount of weight.

To uncover potential risk factors associated with the observed weight gain, we conducted a Spearman’s rank-order correlation test. The results revealed that changes in food intake, emotional eating, and physical activity were all independently and significantly correlated with weight change. Regarding food intake, an increase in the consumption of calorically dense foods, such as chips, potatoes, fries, bread products, cheese, candy, ice cream, cake, and soft drinks, was positively correlated with weight gain. Studies have shown that acute stress can lead to disinhibited eating behaviors, with food choices predominantly favoring sugary and fatty foods [30,33,35,47,48]. Therefore, the stress associated with the coronavirus pandemic, and subsequent school closures, may explain the reported increase in the consumption of these highly palatable, rewarding “comfort” foods.

Due to the school closures and shift to an online classroom, daily routines have changed dramatically. In addition to the adaptation of teaching from home, many teachers were engaged in the homeschooling of their own children. Due to these deviations from their typical daily routines, foods consumed during this three-month period may have largely been shaped by convenience [49,50,51]. According to Locher et al. [52], convenience is another characteristic of “comfort food”. As reported in the New York Times [53], during the pandemic, many people did relax their usual food rules and reached for the comfort foods of their childhood, macaroni and cheese, chips, cookies. The need for quick and easy-to-prepare foods became essential as people tried to squeeze in a meal between Zoom meetings. In addition, the shift to working from home meant food was always readily available. Therefore, the shift to working from home combined with the convenience of easily accessible, palatable, and emotionally comforting foods may further explain some of the weight gain observed among the teachers.

In contrast to the positive association seen between weight gain and the intake of these comfort foods, a negative association was observed between weight gain and the consumption of fresh fruit and beans. Previous studies have illustrated that protein is the most satiating macronutrient, whereas fatty foods are the least satiating. Therefore, due its weaker role of promoting satiation and its high palatability, the increase in the ingestion of fatty foods, such as chips, fries, cheese, and ice cream, demonstrated in this study, may have led to an increase in overall caloric consumption and, consequently, weight gain. Legumes, on the other hand, are low in fat and high in protein and fiber. As with protein-rich foods, fiber, which is found in legumes, as well as fruits, is another contributor to satiety. Therefore, a decrease in the consumption of satiety-inducing beans and fruit may also further explain some of the weight gain observed [54,55,56].

Several studies revealed significantly higher levels of stress reported by teachers during quarantine, due to difficulties adjusting to distance education and the increased workload associated with working from home [57,58]. As previously mentioned, stress is associated with less physical activity and more sedentary behavior, which may contribute to weight gain [40,41,42]. Table 2 reveals a significant decrease in the frequency of both strenuous and moderate exercise, and a significant increase in the frequencies of mild/light exercise and sedentary activities among participants who gained more than 10 pounds when compared to participants who lost more than 10 pounds. Further, an increase in the occurrence of exercise at any intensity (strenuous, moderate, mild/light) was negatively correlated with weight gain, whereas an increase in sedentariness was positively correlated with weight gain. Specifically, we noted that teachers who lost weight (1 to 10+ pounds) had a significant increase in the frequency of strenuous exercise compared to teachers who gained weight (1 to 10+ pounds).

As exhibited in Figure 1B, we saw a significant increase in weight gain among K-5 teachers when compared to teachers of older grade levels. Although several studies showed heightened levels of stress among teachers during school closures [57,58], Ozamiz-Etxebberia et al. [59] reported that the highest levels of stress, and stress-precipitated anxiety, were observed among K-5 teachers. Interestingly, these findings contradict non-quarantine conditions in which high school teachers report being most affected by stress [60]. The increase in stress exhibited by the K-5 teachers may be attributed to the age of the students they teach; younger students (aged 4 to 10) generally require more care and guidance than older students. With schools closed, and face-to-face interaction no longer possible, these teachers may feel that they are no longer able to adequately carry out these duties of care, resulting in heightened stress. As previously mentioned, stress is correlated with an increase in emotional eating, the consumption of comfort foods, and sedentariness; all risk factors for weight gain.

Although teachers of every grade level, spend similar amounts of time dedicated to the education of their students, their daily responsibilities in the classroom greatly differ. Kindergarten and elementary teachers work with children during their first years of school. A significant amount of movement and exploration occur in the classrooms of younger children; teachers need to use a lot of play, games, and hands-on teaching activities to keep 4- to 10-year-old students engaged. In addition, K-5 teachers ambulate frequently throughout the day as they shift between the academic classroom stations, escort students to different activities located throughout the school building, and monitor lunch, recess, and bus dismissal [61,62]. With school closures, days of frequent activity and movement throughout the school building were no longer occurring, perhaps playing a role in the weight gain seen among the K-5 teachers.

Lastly, as previously stated, a significant increase in eating while working was observed among K-5 teachers when compared to MS and HS teachers. Eating while engaged in other activities is an example of mindless eating and, as Wansink [63] has demonstrated, mindless eating can result in a failure to respond to internal satiety cues and, subsequently, causes increased potential of overeating. Therefore, this increase in mindless eating during school closures may provide further explanations in regard to the weight gain detected in this group.

At this point, over 43% of Americans are fully vaccinated [64]. However, with the appearance of new and more contagious COVID variants around the world [65], it seems that future pandemics, and mandatory shelter-in-place regulations, may be an inevitable threat. If these results were extrapolated to the general population, it could result in the adaptation of proactive measures regarding food consumption, emotional eating, and frequency of physical activity during any future quarantines.

## 5. Conclusions

In conclusion, these findings provide the first look at the impact of the COVID-19 school closures on weight and weight-related behaviors among schoolteachers. Weight gain among study participants was independently and significantly associated with increases in the consumption of “comfort” foods, emotional eating and sedentariness. These observed changes in weight-related behaviors among the teachers may have been precipitated by increased stress due to changes in daily schedules, the rapid shift to an online classroom, and worries about the virus itself. Greater levels of stress were reported among kindergarten and elementary (K-5) schoolteachers than middle and high school teachers during the school closures, which may help explain the greater amount of weight gain observed among K-5 teachers. This study provides a starting point for future research looking at the impact of quarantining on behaviors affecting weight. Follow-up studies are needed to investigate whether weight gained during the pandemic was lost once the quarantine restrictions were lifted and to determine if the weight gain was associated with an increase in the incidence of metabolic disorders.

## 6. Limitations

The present study had several caveats. The retrospective design of the study allowed us to estimate associations only and the self-reported questions for weight may have been affected by bias. With the use of social media as the means of recruitment, it could be argued that the study sample was not an adequate representation of the population. In addition, future studies involving a more diverse population would be beneficial. Lastly, it should be mentioned that the existence of unknown confounding factors may have contributed to weight gain (e.g., certain medications. medical conditions). Although these are possible limitations, the current study still provides unique information about the potential impact of quarantine on weight and weight-related risk factors.

## Figures and Tables

**Figure 1 nutrients-13-02723-f001:**
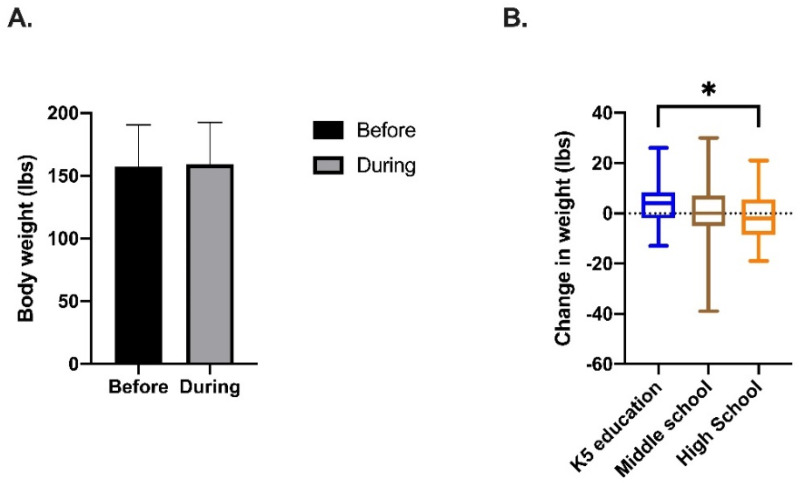
Body weight change (in pounds) for all participants (**A**) and separated by grade taught (**B**). (**A**) reports the mean and standard deviation of body weight for participants before and during the school closures. (**B**) reports Median and IQR. Kruskal-Wallis rank-sum tests were used to determine significance. * indicates *p* value < 0.05.

**Figure 2 nutrients-13-02723-f002:**
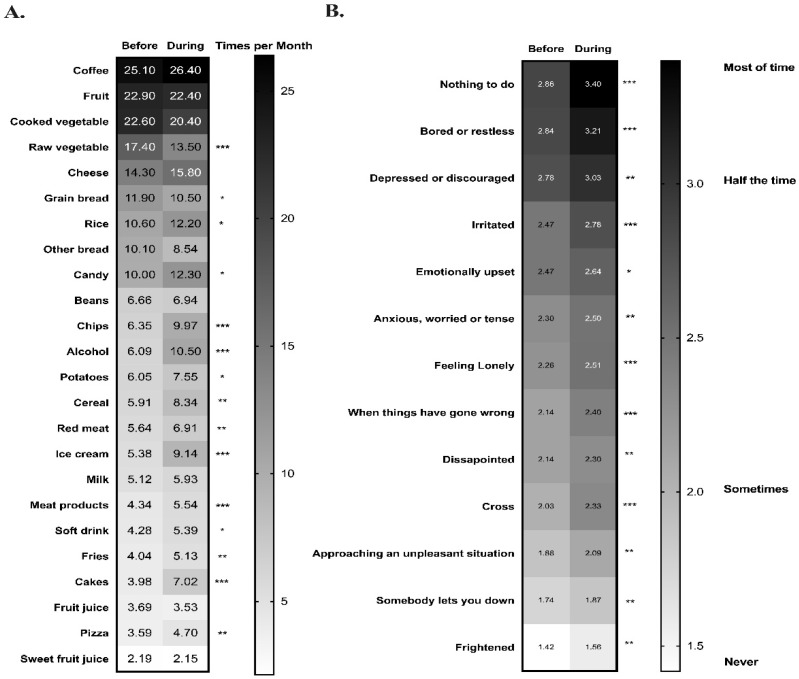
Heat map indicating changes in food consumption and emotional eating. Means of the frequencies of foods consumed (**A**) and emotional eating (**B**) were reported for before school closures (before 16 March) and during school closures (until 26 June). Darker colored cells represent higher frequency of the parameter measured. Wilcoxon signed-rank tests were used, and *, **, *** indicating *p*-value < 0.05, 0.005, 0.0005, respectively.

**Table 1 nutrients-13-02723-t001:** Baseline characteristics of participants.

	K5 Education	Middle School	High School	All
Grade teaching	66 (51%)	34 (26%)	29 (23%)	129
Year teaching				
Below 10 years	19 (29%)	11 (32%)	8 (28%)	38 (30%)
Between 10 to 20 years	20 (30%)	16 (47%)	12 (41%)	48 (37%)
More than 20 years	27 (41%)	7 (21%)	9 (31%)	43 (33%)
Gender				
Female	62 (94%)	32 (94%)	22 (76%)	116 (90%)
Male	4 (6%)	2 (6%)	5 (17%)	11 (9%)
Prefer not to respond	0 (0%)	0 (0%)	2 (7%)	2 (2%)
Age group (year)				
20–29	8 (12%)	8 (24%)	2 (7%)	18 (14%)
30–39	11 (17%)	9 (27%)	9 (31%)	29 (23%)
40–49	28 (42%)	12 (35%)	9 (31%)	49 (38%)
50–59	15 (23%)	5 (15%)	6 (21%)	26 (20%)
60+	4 (6%)	0 (0%)	3 (10%)	7 (5%)
Race				
White	59 (89%)	31 (91%)	25 (86%)	115 (89%)
Asian	2 (3%)	0 (0%)	2 (7%)	4 (3%)
Black or African American	1 (2%)	0 (0%)	0 (0%)	1 (1%)
Other or prefer not to respond	4 (6%)	3 (9%)	2 (7%)	9 (7%)
Change in body weight (lbs.)				
Mean	3.7	0.2	−1.2	1.7
SD	8	11.3	9	9.4
Household income (USD)				
below USD 100,000	13 (20%)	4 (12%)	7 (24%)	24 (19%)
between USD 100,000 and USD 150,000	15 (23%)	9 (27%)	4 (14%)	28 (22%)
above USD 150,000	31 (46%)	19 (56%)	16 (55%)	65 (50%)
Prefer not to respond	8 (12%)	2 (6%)	2 (7%)	12 (9%)

**Table 2 nutrients-13-02723-t002:** Factors affecting weight gain. Spearman’s rank-order correlation test were used to measure significance.

Change in Food Intake	Correlation Coefficient	*p*-Value
Fresh fruit	−0.17	0.049 *
Chips	0.25	0.001 **
Potatoes	0.2	0.03 *
French fries, home fries, or hash browns	0.22	0.01 *
Other bread products	0.21	0.02 *
Cheese (do not include cheese on pizza)	0.18	0.02 *
Beans or pulses (lentils, green peas)	−0.26	0.003 **
Candy or chocolate (not sugar-free)	0.25	0.004 **
Ice cream or other frozen desserts	0.29	0.005 *
Cakes, scones, pies, pastries, biscuits, brownies	0.24	0.04 *
Soft drinks (not diet)	0.19	0.0005 ***
Change in Physical Activity		
Strenuous exercise	−0.44	0.0005 ***
Moderate exercise	−0.31	0.0005 ***
Mild/Light exercise	−0.18	0.04 *
Sedentary activities	0.26	0.003 **
Change in Emotional Eating		
Irritated	0.33	0.0005 ***
Depressed or discouraged	0.18	0.047 *
Approaching an unpleasant situation	0.22	0.01 *
Things have gone wrong	0.22	0.01 *
Anxious, worried, or tense	0.28	0.01 *
Emotionally upset	0.25	0.005 *
Nothing to do	0.22	0.01 *
Feeling lonely	0.2	0.02 *
Bored or restless	0.22	0.01 *

*, **, *** indicating *p*-value < 0.05, 0.005, 0.0005, respectively.

**Table 3 nutrients-13-02723-t003:** Changes in food consumption and exercise intensity according to weight change (pounds). One-way ANOVA rank tests were used. Significant changes are represented by non-overlapping letters across the rows.

	Lost > 10 Pounds	Lost < 10 Pounds	No Change	Gain < 10 Pounds	Gain > 10 Pounds
**Food intake changes**					
Cereal	0 (0–8.7) a	0 (0–9.5) a	0 (−0.4–0) b	0 (0–1.5) ab	0 (0–8) ab
Chips	0 (−1.5–0.8) a	0 (0–1.5) a	0 (0–1.9) ab	1.5 (0–8.4) b	8 (0–9.5) b
Potatoes	0 (−0.7–5.5) a	0 (0–0) a	0 (0–1.5) a	0 (0–0.4) a	1.5 (0–10) b
Fries	0 (−0.8–0) a	0 (0–1.5) ab	0 (0–1.5) ab	0 (0–1.5) b	0 (0–8) b
Other Breads	0 (−10.5–0) a	0 (−8–0) a	0 (−1.9–1.9) ab	0 (0.4–1.9) b	0 (0–5.5) b
Candies	0 (−4.8–1.5) a	0 (−5.5–1.5) a	0 (0–8.5) ab	0 (−0.4–8) ab	0 (0–10.5) b
Ice cream	0 (−1.5–1.5) a	0 (0–8) ab	0 (0–3) abc	0.8 (0–8) bc	1.5 (0–9.8) c
Cakes	0 (−1.5–0.8) a	0 (0–5.5) ab	0 (0–1.5) ab	0 (0–8) ab	0 (0–10.5) b
Soft drinks	0 (0–0) a	0 (0–0) a	0 (0–0) a	0 (0–0) a	0 (0–5.5) b
**Physical activity changes**					
Strenuous exercise	3 (0.5–5) a	1 (0–3) ab	0 (−0.3–1) bc	0 (−0.3–0) c	0 (−1.5–0) c
Moderate exercise	3 (0–4) a	2 (0–3) a	1 (0.8–2) a	0 (−2–3) ab	−1 (−3–1.5) b
Mild/light exercise	1 (0–3) a	0 (0–0.5) bc	0 (0–3.3) ab	0 (0–1) abc	0 (0–1) c
Sedentary activities	2 (−0.5–5) a	3 (1.5–6) ab	2.5 (2–5) ab	3 (2–5) ab	5 (4–6) b

**Table 4 nutrients-13-02723-t004:** Changes in foods consumed by grade taught. One-way ANOVA rank tests were used. Significant changes are represented by non-overlapping letters across the rows.

	K5 Education	Middle School	High School
Fruits	0 (−9.5–0) a	0 (0–9.9) b	0 (0–8) b
Raw Vegetable	−8 (−9.9–0) a	0 (−6.8–1.1) b	0 (−8–0) ab
Chips	1.5 (0–9.5) a	0 (0–2.6) ab	0 (−1.5–1.5) b
Meat-products	0 (0–3) a	0 (0–1.1) ab	0 (0–0) b

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
