# Peer review of "Impact of School Closures, Precipitated by COVID-19, on Weight and Weight-Related Risk Factors among Schoolteachers: A Cross-Sectional Study"

_nutrients, 2021, doi:10.3390/nu13082723_

Round 1

Reviewer 1 Report

Silverman et al. analyzed the impact of the COVID-19 and school closures on weight and weight-related risk factors among schoolteachers with a cross-sectional study. Once subjects voluntarily agreed to participate in the study, they received the link to a 87 Qualtrics (Seattle, USA) survey consisting of 52 questions. The authors found that 52% of study participants reported weight gain, with a higher prevalence observed among kindergarten and elementary school teachers when compared to high school teachers (p < 0.05). Weight gain observed among teachers during the school closures, was associated with changes in diet, emotional eating and physical activity.

The article is well-written and overall the results are presented clearly. 

I have some comments:

1) The reason why kindergarten and elementary school teachers reported higher rates of weight gain when compared to high school teacher should be further analyzed since it represents one of the crucial messages of this manuscript.

2) A limitation section should be included in the manuscript, including all missing variables/data that this kind of questionnaire may had in its design. For example, COVID-19 lockdown has resulted in higher rates of depression or other psychiatric disorders. A higher use of antidepressants might represent an interesting variable that might also have contributed to a significant weight gain.

3) Did the authors collect data on Metabolic syndrome as well? Data so far published point towards metabolic diseases (mainly diabetes mellitus and obesity) as conditions associated with more severe outcomes, including higher mortality risk, in the setting of COVID-19. Surely, a significant weight gain may result in a long-term worsening clinical picture on new-onset COVID-19 case. Moreover, although obesity has been reported as a major risk factor for severe outcomes, the role of dyslipidaemia and of the subsequent statin use in COVID-19 is so far controversial. In some studies the use of statin has not been associated with favorable outcomes in COVID-19: “Impact of prior statin use on clinical outcomes in COVID-19 patients: data from tertiary referral hospitals during COVID-19 pandemic in Italy. J Clin Lipidol. 2021 Jan-Feb;15(1):68-78. doi: 10.1016/j.jacl.2020.12.008.”, while in other reports a potential improvement has been suggested “Statin therapy in COVID-19 infection. Eur Heart J Cardiovasc Pharmacother. 2020 Jul 1;6(4):258-259. doi: 10.1093/ehjcvp/pvaa042.” – “Statin therapy in COVID-19 infection: much more than a single pathway. Eur Heart J Cardiovasc Pharmacother. 2020 Nov 1;6(6):410-411. doi: 10.1093/ehjcvp/pvaa055.” Please comment on the role of MetS and eventually on dyslipidemia and  statin therapy in this specific clinical setting.

Author Response

Thank you for taking the time to review this manuscript and for your constructive feedback.

Point 1: The reason why kindergarten and elementary school teachers reported higher rates of weight gain when compared to high school teacher should be further analyzed since it represents one of the crucial messages of this manuscript.

A new paragraph was added in the discussion section to further support our findings. Previous studies demonstrate that K-5 teachers experienced higher levels of stress teaching from home than in a classroom setting. Conversely, upper level school teachers usually experience more stress in the classroom setting. Increased levels of stress are associated with emotional eating, consumption of comfort foods and sedentariness which are all risk factors for weight gain.

Point 2: A limitation section should be included in the manuscript, including all missing variables/data that this kind of questionnaire may had in its design. For example, COVID-19 lockdown has resulted in higher rates of depression or other psychiatric disorders. A higher use of antidepressants might represent an interesting variable that might also have contributed to a significant weight gain.

A limitations section was added to the manuscript explaining potential caveats such as method of subject recruitment, bias of self-reported weights, and other potential explanations for the observed weight gain (e.g. medications, medical reasons).

Did the authors collect data on Metabolic syndrome as well? Data so far published point towards metabolic diseases (mainly diabetes mellitus and obesity) as conditions associated with more severe outcomes, including higher mortality risk, in the setting of COVID-19. Surely, a significant weight gain may result in a long-term worsening clinical picture on new-onset COVID-19 case. Moreover, although obesity has been reported as a major risk factor for severe outcomes, the role of dyslipidaemia and of the subsequent statin use in COVID-19 is so far controversial. In some studies the use of statin has not been associated with favorable outcomes in COVID-19: “Impact of prior statin use on clinical outcomes in COVID-19 patients: data from tertiary referral hospitals during COVID-19 pandemic in Italy. J Clin Lipidol. 2021 Jan-Feb;15(1):68-78. doi: 10.1016/j.jacl.2020.12.008.”, while in other reports a potential improvement has been suggested “Statin therapy in COVID-19 infection. Eur Heart J Cardiovasc Pharmacother. 2020 Jul 1;6(4):258-259. doi: 10.1093/ehjcvp/pvaa042.” – “Statin therapy in COVID-19 infection: much more than a single pathway. Eur Heart J Cardiovasc Pharmacother. 2020 Nov 1;6(6):410-411. doi: 10.1093/ehjcvp/pvaa055.” Please comment on the role of MetS and eventually on dyslipidemia and  statin therapy in this specific clinical setting.

Although we did not collect data on metabolic disorders, we appreciate the reviewer's suggestion and expanded a sentence in the discussion section highlighting the importance of including this topic in future studies:

"Follow up studies are needed to investigate whether weight gained during the pandemic was lost once the quarantine restrictions were lifted and to determine if the weight gain was associated with an increase in the incidence of metabolic disorders."

Reviewer 2 Report

The problem of remote education in times of the covid-19 epidemic is a serious problem. The authors undertook to investigate the effect of forced lifestyle change on body weight.

line 89 - there is no description of how the authors of the study obtained values ​​from anthropometric measurements that were used to assess the change in body weight.
Was BMI and its change calculated or only body weight? if so, why was the height of the respondents in the questionnaire? did the respondents receive guidelines on how to make measurements?

Gender and race described in Table 1 are not described in more detail, i.e. the very important question of how race influenced eating behavior during the pandemic, and whether women or men were more likely to experience poor eating habits, emotional eating, and increased body weight. This should be completed.

line 299-301 - this fragment is not related to the work, I would suggest removing it.
There is a lack of strength and limitation and no conclusions that can be drawn from the research. these sections must be completed.

Author Response

Thank you for taking the time to review the manuscript and for your  constructive feedback.

Point 1: line 89 - there is no description of how the authors of the study obtained values ​​from anthropometric measurements that were used to assess the change in body weight.
Was BMI and its change calculated or only body weight? if so, why was the height of the respondents in the questionnaire? did the respondents receive guidelines on how to make measurements?

Since the goal of this study was to determine weight change, rather than degree of overweight, we used the teachers' self-reported weights for all analyses; height was not used in any calculations. There were no specific guidelines in regard to the weight measurements which is addressed in the limitations section of the manuscript.

Point 2: Gender and race described in Table 1 are not described in more detail, i.e. the very important question of how race influenced eating behavior during the pandemic, and whether women or men were more likely to experience poor eating habits, emotional eating, and increased body weight. This should be completed.

Although, we used several Facebook teacher pages to promote participation in the study, most respondents were white females. Even though this is reflective of the national demographics (80% white and 77% female), the authors agree that future studies should include greater diversity among participants to determine differences in regard to race and gender. We added a sentence in the limitations section to address this point:

"In addition, future studies involving a more diverse population would be beneficial" 

Point 3: line 299-301 - this fragment is not related to the work, I would suggest removing it.
There is a lack of strength and limitation and no conclusions that can be drawn from the research. these sections must be completed.

Author unclear as to which fragment reviewer refers. Eating while working is an example of mindless eating which is associated with increased food intake and, subsequently, and increase in weight. The K-5 teachers reported an increase in eating while working (mindless eating) during quarantine which may provide explanation in regard to the weight gain observed among this group. Lines 299-301 support those results. Hopefully, this adequately explains the insertion of those lines in the manuscript.

Round 2

Reviewer 1 Report

I think the manuscript has been sufficiently improved and can be considered for publication.

Author Response

Thank you again for reviewing the manuscript and for the constructive feedback - it is greatly appreciated.

Reviewer 2 Report

the work has been corrected and the authors' explanations are perfectly correct. The limitation section is very much needed, well it has been added to the work. However, the separate section Conclusion is missing all the time and in my opinion the authors should supplement / re-edit it. 

Author Response

Thank you again for taking the time to review the manuscript and for the constructive feedback.

A conclusion section has been added, summarizing the main findings and how they can be used to develop this subject further in future studies.